# Myths about Sexual Aggression, Sexual Assertiveness and Sexual Violence in Adolescent Romantic Relationships

**DOI:** 10.3390/ijerph17238744

**Published:** 2020-11-25

**Authors:** Andrés A. Fernández-Fuertes, Noelia Fernández-Rouco, Susana Lázaro-Visa, Eva Gómez-Pérez

**Affiliations:** Department of Education, University of Cantabria, 39005 Santander, Spain; fernandezrn@unican.es (N.F.-R.); lazaros@unican.es (S.L.-V.); eva.gomez@unican.es (E.G.-P.)

**Keywords:** adolescence, sexual violence, assertiveness, myths, dating violence

## Abstract

Sexual violence is a worldwide health and social issue. However, little is known about the sexual violence that occurs in the context of romantic relationships. This study analyzes the existence of sexual violence in adolescents’ romantic relationships, the possible associations between such violence (both committed and suffered) and myths about sexual aggression and sexual assertiveness, and the possible gender-related distinctions. A sample of 329 students aged between 15 and 19 (M = 16.3; SD = 0.79) was surveyed; all participants were in a romantic relationship when the data were collected. The results reveal that both genders report the existence of sexual violence in their romantic relationships; however, in heterosexual relationships, males were more often the perpetrators of sexual violence. Additionally, myths about sexual aggression and sexual assertiveness were found to be significantly associated with both perpetration and victimization. Accordingly, these predictors should be focal points in prevention programs for adolescent sexual violence. The findings of this study show the de-prioritization of sexuality education in the Spanish educational system, as well as the need to strengthen the competence of adolescents in this area.

## 1. Introduction

Human sexuality is rich and diverse because it is not biologically determined; as a result, it is possible to decide when, how and with whom to be intimate, provided that the individuals involved give voluntary and free consent [1]. Sexual encounters, however, are not always consensual: coercion and sexual violence can have significant repercussions on physical and psychological health, and can even impact the long term ability to form healthy intimate relationships [2,3].

Sexual violence can be defined as “a sexual act that is committed or attempted by another person without freely given consent of the victim or against someone who is unable to consent or refuse” (p. 11) [4]. There are different ways in which a person’s sexual freedom can be violated, from the use of physical force or the threat of doing so, to emotional manipulation and alcohol and/or drug incapacitation [5]. Moreover, there are a number of different forms of sexual violence (e.g., kissing, fondling, masturbation, oral sex, coital sexual violence, etc.); though it is not the most common form of sexual violence, most studies have focused on the analysis of coital sexual violence [6,7]. A meta-analytic review of the prevalence rates of dating violence among adolescents concluded that approximately 1 in 10 adolescents reported experiencing sexual violence [8]; however, systematic bias in participants’ self-reports was evident, as was high variability in the rates of dating violence among studies. Most research on sexual violence in adolescent and young adult relationships has been conducted in the United States [9]. A small number of studies have been carried out in Spain, although generally with samples of university students, despite the fact that the first sexual and romantic relations are often established before the age of 18 [10]. The work of Sipsma, Carrobles, Montorio, and Everaerd was one of the first studies on sexual violence among young adults in Spain: in their research, 33.2% of female university students surveyed reported having been victims of one or more acts of sexual violence by peers [11]. More recently, Romero-Sánchez and Megías found that approximately 36% of female university students had been victims of sexual violence and that 16% of male university students acknowledged having engaged in sexual violence [12]. Another study focused on adolescent relationships found that about 57% of adolescent females and 58% of adolescent males had experienced some form of sexual violence at the hands of their partners (e.g., kissing, touching, undressing, sexual relations, etc. without consent), figures which are comparable to the percentage of adolescent females and males who admitted to having perpetrated these behaviors: 44% and 61%, respectively [6]. These studies reveal that, while both young men and women may commit and be subjected to sexual violence, young men tend to perpetrate it more frequently and, consequently, young women are more often the victims of such acts. This conclusion is also supported by various international studies [13,14]: this is arguably why most studies have focused on the analysis of sexual violence committed by males and/or sustained by females [9,15,16].

Violence in dating relationships is associated with a variety of adverse effects, including lowered self-esteem, reduced self-worth, increased self-blame, anger, hurt, and anxiety. Moreover, teenage dating violence victimization increases the risk of physical injury, antisocial behaviors, suicidal ideation, and continued violence in adult relationships [17,18]. Studies have also documented the long-term impact of dating violence on survivors, which includes prolonged isolation, the withholding of emotional support, and an increased likelihood of continued experiences of abuse, such as harassment and degradation [19,20].

A number of interventions have been developed to address this problem, but most have been created using theory and conclusions drawn from the literature on adult partner violence, in part because of the paucity of rigorous longitudinal studies on young people [21,22]. Unlike a simple bivariate relationship, sexual violence is the result of a chain of events, and therefore it is necessary not only to consider different predictor variables but also the different relationships that exist among them (mediation, moderation, etc.) for the promotion of behavioral change [8,23,24]. Several theoretical approaches aim to both understand and explain the violence perpetration and victimization. For example, the Confluence Model explains sexual aggression on the basis of developmental, personality, and behavioral elements that have been shown to be associated with this behavior (i.e., hostile masculinity and impersonal sex), particularly among male perpetrators in the general population [25]. Alternatively, the Theory of Behavior Change considers that greater knowledge and stronger interpersonal skills are linked to a decrease in dating violence [23]. There is no doubt that understanding those factors associated with sexual violence is important. Myths about gender, rape and other forms of sexual violence are among the most widely analyzed factors at an international level [16,26]. In this regard, some studies have observed that the rate of adherence to such false beliefs (e.g., “women lie about rape”; “men cannot control their behavior when they are very excited”, etc.) is often higher among those who commit sexual violence, most commonly young men [13,27,28]. However, the explanatory power of these false beliefs among young men and women has scarcely been studied, particularly when analyzing sexual violence committed and suffered by both genders in romantic relationships. Evidence from dating violence prevention programs shows that educating participants by dispelling false beliefs and romantic myths, and shifting attitudes away from supporting violence in romantic relationships is beneficial for all youth [18,29,30,31], especially if they have sufficient duration (long-term programs) and take place in a variety of contexts (e.g., school, family, community, etc.) [18,26,32,33].

A lack of positive communication and consent negotiation skills also appears to be linked to dating violence [26,29]. As such, a more developed set of interpersonal skills would likely lead to both a reduction in the onset of dating violence, as well as a decline in the perpetration and victimization of the same. Behavior change, therefore, tends to emerge not only from changes in attitude and increased awareness, but also as a result of tangible skill development [18,33,34]. Tharp et al. conclude that, in romantic relationships, poor negotiating skills are associated with the implementation of coercive strategies when trying to fulfill the need for intimacy [16]; moreover, according to Malamuth, Feshbach and Jaffe, sexual violence against women might reduce a perpetrator’s anxieties about being rejected by eliminating the victim’s ability to exercise choice [35]. Sexual assertiveness is defined as the ability to act independently with regard to one’s own sexuality [36]. Walker, Messman-Moore and Ward found that sexual assertiveness might act as a protective factor against sexual victimization [37], a conclusion also supported by a study conducted in Spain [27]. Little analysis, however, of the possible relationships between sexual assertiveness and sexual violence perpetration has been carried out. In addition, when the link between sexual assertiveness and sexual violence has been explored, it has frequently been limited to sexual violence suffered by young adult women, with no consideration of adolescents or male victims.

With regard to decreasing dating violence among adolescents, there is hope that greater knowledge of dating violence and improved skill development will effectively prevent or reduce incidents of such violence [18,34]. Taking into account prior international literature and given that most studies have been largely limited to exploring the existence of peer assault, especially that committed by males and/or experienced by females [15,38], the first objective of this study is to obtain data on sexual violence committed and sustained by adolescents, both male and female, in the context of romantic relationships. Because it has not been adequately studied internationally or in Spain, the second objective is to explore the explanatory power of two constructs (i.e., myths about sexual aggression and sexual assertiveness) in relation to sexual violence (both committed and suffered) by male and female adolescents in romantic relationships.

## 2. Materials and Methods

### 2.1. Participants

The initial study sample consisted of 930 participants from Cantabria (Spain): participants were solicited via nine high schools randomly selected from 20 high schools that had previously signed up to participate in a sexual violence prevention program. Sampling was carried out in clusters in randomly selected classrooms at each high school. Families and adolescents had previously consented to participate in both the program and the study: before signing the consent form (99.5% report rate), all participants were informed of the main objectives and characteristics of the program and the study, as well as its voluntary nature. The following eligibility criteria were established a posteriori: first, questionnaires with response omissions under 20% (11 participants excluded); second, being in the 15–19 year age range (22 participants excluded); and finally, being in an established romantic relationship of one month or more at the time of participation in the study (551 participants excluded). In addition, given that the number of participants who self-identified as non-heterosexual and/or non-binary gender was too small for comparisons, these participants were not taken into account in this study (17 participants excluded).

The final sample was composed of 329 adolescents, 190 males and 139 females. Their mean age was 16.31 (SD = 0.79): no statistically significant differences were found between males (M = 16.21; SD = 0.68) and females (M = 16.39; SD = 0.91) in this variable. However, statistically significant differences were obtained between genders in the mean age of their current partner (M = 16.23; SD = 1.17): the results of the one-way between-groups ANOVA (F(1,327) = 30.82; *p* < 0.001) indicated that females’ partners (M = 16.63; SD = 1.18) were older than males’ partners (M = 15.93; SD = 1.07), even though the actual difference in mean scores between both groups was not large (ŋ^2^ = 0.08). Another one-way between-groups ANOVA was conducted to explore gender differences in terms of the duration of the current romantic relationship, a variable that was evaluated in months (M = 5.14; SD = 6.81): it was found that females (M = 6.37; SD = 7.91) reported having longer relationships than males (M = 4.25; SD = 5.61); however, despite reaching statistical difference (F(1,327) = 7.92; *p* < 0.01), the effect size was small (ŋ^2^ = 0.02).

### 2.2. Procedure

Data collection, carried out throughout the ten weeks leading up to the start of the aforementioned prevention program, took place during school hours in a single session. In an effort to elicit truthful responses, teachers were asked to leave the classroom during the application of the questionnaire. To ensure that the anonymous questionnaire was clearly understood and that privacy was maintained, at least one member of the research group was present at each session.

The work was developed in accordance with international and national ethical recommendations, drawing on the Code of Good Practice of the Spanish National Research Council [39], and all procedures were approved by the University ethics committee. In all phases of data collection, the voluntary and confidential nature of the study was emphasized. Additionally, participants were informed that they could decline to participate at any time and that filling in the questionnaire was not mandatory. In order to reconfirm informed consent, before beginning data collection, participants were reminded of the main characteristics of the study, as had been done previously with them and their families. Participants were informed that by returning the completed questionnaire to the researchers, they were reconfirming the previously given informed consent. In addition, the most relevant information regarding the research appeared on the first page of the questionnaire. The order of the instruments was as follows: socio-demographic data, myths about sexual aggression, sexual assertiveness, and sexual violence (both committed and suffered). Additionally, before presenting the different forms of sexual violence that may have taken place, participants were asked whether they had ever been involved as the agent or recipient in an unwanted sexual situation with their current partner, which served to emphasize the unwanted nature of the list of behaviors that followed.

### 2.3. Measures

Sexual violence committed: The Conflict in Adolescent Dating Relationships Inventory (CADRI) assesses the occurrence of five forms of violence among teen dating partners: verbal or emotional abuse (10 items; e.g., “I insulted her/him with put downs”), physical abuse (4 items; e.g., “I kicked, hit, or punched her/him”), threatening behavior (4 items; e.g., “I deliberately tried to frighten her/him”), relational aggression (3 items; e.g., “I spread rumors about her/him”) and sexual abuse (4 items; e.g., “I kissed her/him when s/he didn’t want me to”) [40]. Given the aim of this study, only the sexual abuse subscale was used: specifically, the subscale that is part of the revised Spanish version of the CADRI [41], which incorporates two additional offenses beyond the four types of sexual violence included in the original instrument (i.e., “I forced my partner to touch my breasts, genitals and/or buttocks when s/he didn’t want to”, and “I kept removing my partner’s clothes even though I knew s/he didn’t want me to”). Participants were asked to complete this instrument only in reference to their current dating relationship. The response modality consists of a rank scale of four options ranging from 0 (“Never: this has never happened in your relationship”) to 3 (“Often: this has happened 6 times or more in your relationship”). The alpha of Cronbach obtained in this study was 0.76 (6 items).

Sexual violence suffered: The CADRI is made up of bidirectional items which measure how often participants were involved in violent behavior, both in perpetration (e.g., “I touched her/him sexually when s/he didn’t want me to”) and in victimization (e.g., “S/he touched me sexually when I didn’t want her/him to”) [40]. In this study, only the sexual abuse subscale that is part of the revised Spanish version of CADRI was used. It was applied to detect sexual violence suffered in participants’ current romantic relationships [41]. The reliability observed was 0.75 (6 items).

Myths about sexual aggression: The Acceptance of Modern Myths about Sexual Aggression Scale (AMMSA) is a one-dimensional, 30-element self-report that analyses different myths about sexual aggression (e.g., “Many women tend to exaggerate the problem of male violence”) [42]. The response format is a seven-point Likert scale, ranging from 1 (“Completely disagree”) to 7 (“Completely agree”). In this work, an internal consistency of 0.87 was obtained, in accordance with what was observed in a previous study of university students in Spain [43].

Sexual assertiveness: The Spanish version of the Sexual Assertiveness Scale (SAS) was used [44,45]. The 18 items were divided into three categories: Initiation (e.g., “I initiate sex with my partner if I want to”; 6 items), Refusal (e.g., “I refuse to let my partner touch my breasts if I don’t want him/her to, even if my partner insists”; 6 items) and Pregnancy and Sexually Transmitted Infections (STI) Prevention (e.g., “I refuse to have sex if my partner refuses to wear a condom or latex barrier”; 6 items). The response modality consists of a scale composed of five options: from 0 (“Never, 0% of the time”) to 4 (“Always, 100% of the time”). The internal consistency obtained in the subscales was as follows: 0.72 for Initiation, 0.74 for Refusal, and 0.82 for Pregnancy and STI Prevention.

### 2.4. Data Analysis

Coding and data analysis was performed for a significance level of 0.05. First, the appropriateness of the data was examined and only those questionnaires with response omissions under 20% were considered. Second, descriptive analyses of the variables under study were carried out. Third, one-way between-groups analysis of variance (ANOVA) was used to detect statistically significant differences between genders on main sociodemographic variables and eta squared (ŋ^2^) to calculate the effect size when necessary. Fourth, multivariate analysis of variance (MANOVA) was carried out to compare both genders in terms of their means on the variables analyzed (i.e., sexual violence committed, sexual violence suffered, myths about sexual aggression and sexual assertiveness). In the comparisons made, the *F* offered by Wilks’ Lambda was used to interpret the significance of the model and partial eta squared (ŋ^2^*_p_*) to identify the effect size [46]. Finally, Pearson correlation and hierarchical multiple regression analyses were carried out for the purpose of exploring possible relationships among variables.

## 3. Results

### 3.1. The Exploration of Myths about Sexual Aggression, Sexual Assertiveness, and Sexual Violence

The study of adolescent knowledge reflects that some myths about sexual aggression are still prevalent today. In this regard, over fifty percent of males (60.61%) and females (66.73%) affirmed that it is common for women to complain of sexual assault for no reason other than to feel liberated. In addition, both genders stated, at close rates (71.52% of males and 69.53% of females), that women choose to act coy with regard to whether or not they want to have sex. Furthermore, a significant number of adolescents (63.33% of males and 65.02% of females) responded that victims of sexual violence receive appropriate assistance in the form of women’s shelters, counseling opportunities, and support groups.

With regard to sexual assertiveness, it was found, for example, that 27.73% of males and 16.32% of females admitted to not initiating sex with their partners when they desire it. Similarly, a relevant number of participants reported that they do not refuse caresses despite not wanting them (31.34% of males and 19.02% of females), and that they engage in sexual intercourse even though they do not feel like it (26.11% of males and 22.61% of females). In addition, they never indicate when they would like to be touched (46.64% of males and 43.71% of females).

With respect to sexual violence committed, 43.61% of males and 17.42% of females indicated that they have found themselves in at least one situation in which they have felt like having some kind of sexual contact (e.g., kisses, hugs, caresses, sexual relations, etc.), but their partner clearly stated that he or she did not want it. Among those who found themselves in that situation, an alarming number of participants (51.31% of males and 24.61% of females) admitted to having insisted or pressured their partner in some way, on at least one occasion, in order to attain sexual interaction. Of all the types of violent sexual behavior analyzed, kissing was the behavior indicated to have been performed most frequently by both genders (carried out at least once by 41.62% of males and 19.32% of females), followed by genital caressing (carried out by 34.53% of males and 11.11% of females). It should be noted, however, that the average frequency of sexual violence was low (Table 1), meaning that on the whole, the instances of sexual violence appear to be isolated events.

With respect to sexual violence suffered, 34.54% of males and 30.12% of females indicated that they had been in a situation in which their partner insisted or pressured them to engage in sexual behavior despite the fact that the participant clearly stated that he or she did not want to. Of these people, 57.91% of males and 56.54% of females acknowledged that they finally gave in to the pressure of their partners. Although the average frequency of sexual violence suffered was low (Table 2), when taking into account the different frequency categories collectively, there was a substantial number of participants who reported having been on the receiving end of sexual aggression in their relationships. Nonconsensual kissing, in particular, which appears to be the most common form of sexual violence, was sustained at least once by at least 40.13% of males and 34.34% of females. After kissing, the most frequently suffered aggression was genital caressing, experienced by 23.12% of males and 35.13% of females in their current romantic relationships.

The results of the MANOVA showed significant differences between genders in certain variables. Differences were found in myths about sexual aggression (F(1,306) = 16.53; *p* = 0.000; ŋ^2^*_p_* = 0.051), refusal assertiveness (F(1,306) = 0 47.44; *p* = 0.000; ŋ^2^*_p_* = 0.134), pregnancy and sexually transmitted infections (STI) prevention assertiveness (F(1,306) = 6.78; *p* < 0.05; ŋ^2^*_p_* = 0.022) and sexual violence committed (F(1,306) = 21.27; *p* = 0.000; ŋ^2^*_p_* = 0.065). These results show that females obtained higher scores in two of the assertiveness subscales (i.e., refusal, and pregnancy and STI prevention); while males presented a higher degree of belief in myths about sexual aggression, and reported having committed a higher number of sexual aggressions (Table 1).

### 3.2. Associations Among Myths about Sexual Aggression, Sexual Assertiveness, and Sexual Violence

In order to determine possible relationships among studied variables, a bivariate correlational analysis was carried out. This analysis revealed significant correlations between certain variables. 

In both genders, negative correlations were observed between myths about sexual aggression and sexual assertiveness (Table 2); however, in the case of females, significant correlations were only established between such myths and two forms of assertiveness (i.e., initiation, and pregnancy and STI prevention), while in males a only significant correlation was found between myths and a form of assertiveness (i.e., refusal).

The relationship between sexual violence (both committed and suffered) and myths about sexual aggression was also investigated. The correlations obtained were negative; however, the only statistically significant correlation observed was the positive correlation between myths and sexual violence suffered in female participants (Table 2).

With respect to the relationships between sexual violence and sexual assertiveness, two negative significant correlations were found for both males and females: a correlation between sexual violence committed and refusal assertiveness, and a correlation between sexual violence suffered and pregnancy and STI prevention assertiveness (Table 2). Moreover, some significant correlations were found to be gender-specific: for males, negative correlations were found between sexual violence committed and pregnancy and STI prevention assertiveness, as well as sexual violence suffered and refusal assertiveness; for females, sexual violence suffered was negatively correlated with initiation assertiveness (Table 2).

For male participants, the most pronounced relationship among all the variables studied was between sexual violence committed and sexual violence suffered in the romantic relationship; in the case of females, however, the most marked relationship was that between refusal assertiveness and pregnancy and STI prevention assertiveness (Table 2).

In order to identify which variables might help to predict having suffered or committed sexual violence, while testing the possible moderating effect of gender, hierarchical multiple regression analyses were conducted. Variables that correlated significantly (*p* ≤ 0.05) with sexual violence committed and gender were entered in the first block of the regression analysis (main effects), and interactions between gender and those predictors that correlated significantly only for males or females, or those whose direction of association was different for each gender, were entered in the second block (interactions). This analytical strategy was also used with sexual violence suffered as the criterion variable in an attempt to generate models that could be applied to all participants, regardless of gender. Thus, if the contributions of the interactions were not statistically significant enough to account for the criterion variable, only one main effects model was chosen; however, if the percentage of variance explained by the second block of variables was statistically significant, then the significant interactions were incorporated into the final models of sexual violence committed and sexual violence suffered.

In relation to sexual violence committed, gender, myths about sexual aggression, refusal assertiveness, and pregnancy and STI prevention assertiveness were included in the first block of the regression, and pregnancy and STI prevention assertiveness in interaction with gender in the second. With respect to sexual violence suffered, gender, myths about sexual aggression, refusal assertiveness, and pregnancy and STI prevention assertiveness were introduced in the first block of the regression and, in the second, both myths and pregnancy and STI prevention assertiveness in interaction with gender (see Table 2). After the analysis, it was observed that the inclusion of these interactions caused significant variance in the main effects model (R^2^ = 0.024; F(1,96) = 8.86, *p* = 0.003). The model with interactions, therefore, was chosen to account for sexual violence committed within a couple. Moreover, the predictor variables included as main effects appear to have sufficient independence between them (Table 3).

The resulting final model explained 37.5% of the variance of the sexual violence committed variable (F(3305) = 16.594, *p* = 0.000). The model was formed by two dimensions of assertiveness (refusal and pregnancy and STI prevention), in addition to pregnancy and STI prevention assertiveness in interaction with gender (Figure 1).

Regression analyses for sexual violence suffered showed that the inclusion of interactions did not involve a significant percentage of variance in the main effects model (R^2^ = 0.009; F(2,301) = 1.49, *p* = 0.227). For this reason, the main effects model was selected to explain this criterion variable. The predictor variables that were included as main effects, once again, appeared to have an adequate degree of independence relative to each other (Table 4).

Hence, the resulting final model consisted of a single variable (i.e., assertiveness in pregnancy and STI prevention; Figure 2), which explained 24.9% of the variance within the variable of sexual violence suffered (F(1307) = 20.212, *p* = 0.000). This percentage was not significantly different from the percentage of variance explained by the combination of all variables included as predictors (27.9%), which indicated that the remaining predictor variables did not contribute a large percentage of variance to the explanation of sexual violence suffered.

## 4. Discussion

This study aimed to address a gap in the literature by examining the existence of sexual violence in teen dating relationships. To that end, this study explored the possible associations between committing and suffering sexual violence and myths about sexual aggression and sexual assertiveness for male and female adolescents. There are several important findings from this study. First, the results of this research show that sexual violence seems to have a significant presence on adolescent romantic relationships, although the average frequency of sexual violence reported among the participants was relatively low. In addition, male participants more frequently reported having committed sexual violence, whereas there were no statistically significant gender-based differences with regard to sexual violence suffered.

There are several other interesting findings regarding adolescent romantic relationships worth mentioning: first, certain myths about sexual aggression are still commonly believed among adolescents, especially among males; second, the low level of sexual assertiveness of the participants indicates that there is room for improvement, especially among males; and, third, sexual assertiveness is a particularly important construct for understanding the existence of sexual violence in adolescent romantic relationships, specifically pregnancy and sexually transmitted infections (STI) prevention assertiveness concerning both committed and suffered violence, as well as refusal assertiveness for explaining sexual violence committed.

As previously stated, what is known about sexual violence among adolescents and young people is limited [9,13,15], especially within the context of a romantic relationships [7,18]. In recent decades, interest in this problem has been increasing, reflected by the growing number of scientific studies and preventive measures in this area [15,26,47]. Previous research has shown that increased knowledge and improved interpersonal skills are important factors in preventing or reducing dating violence perpetration and victimization [26,29,34]. Nevertheless, this study shows that certain myths related to sexual aggression are still prevalent among Spanish adolescents to this day while sexual assertiveness was found to be lacking. According to various international studies on sexual violence [13,16,27] and the previous framework on dating violence [23,25], these two elements are related to the presence of sexual violence among teens and young adults. The results of this study further affirm that these associations are relevant when accounting for the sexual violence that occurs in adolescent couples as well.

It is necessary to point out that, as found in this study, adolescent males tend to report having committed sexual violence against their sexual and romantic partners more frequently, and, in addition, that male teens tend to adhere more closely to various myths that seem to underlie the problem of sexual violence in teen relationships (e.g., false beliefs about gender and sexual aggression) [13,27]. The fact that myths on this subject still exist today is worrying, not only because of the correlation with the perpetration of sexual violence, but also due to the fact that greater acceptance of these types of myths seems to be associated with a lesser predisposition to try to help in a situation where a sexual assault is observed [13]. Mouilso and Calhoun state that, to some extent, these beliefs could lead to the justification of these acts or to a relaxation of the social norms by which such behavior is sanctioned [48]. Similarly, some authors argue that the existence of these false beliefs on a social level generates added suffering for the victims, which furthermore hinders their recovery [49]. Actions aimed at eradicating these and other myths that affect the experience of sexuality should therefore be a priority for the educational system [28,50,51].

Similarly, judging from the results of this study, the promotion of sexual assertiveness in both genders should be prioritized as an area of intervention in adolescence. In fact, given that sexual assertiveness appears to have greater explanatory power, it should be prioritized above deconstructing gender-based myths. On this point, in Shafer, Ortiz, Thompson, and Hemmer’s work with male university students, assertiveness was found to be more relevant than the acceptance of rape myths in predicting attitudes, intentions, and interpretation of sexual consent [52]. This likely has to do with the greater connection between this construct and sexual violence, although additional studies would be needed to confirm this view.

As for future lines of work, the use of longitudinal designs based on a mixed methodology (i.e., quantitative and qualitative) would be advantageous. In addition, by investigating the effects of mediation and moderation among the predictors being studied, the effect of each of the variables on other possible risk factors could be determined. Likewise, it would be equally valuable to work directly with a sample of couples, rather than using participants who self-report on their own and their partner’s behavior. Such an approach would contribute to an understanding of why significant gender-based differences exist with regard to the frequency of violence committed, but not the frequency of violence suffered. Furthermore, it would make it possible to analyze the extent to which the members of the couple agree on what had happened in the relationship, an aspect that has proven useful in previous studies [53]. In addition, other possible influences on the origin and perpetuation of this problem, such as family and peers, must be explored further. Moreover, it is necessary to study how adolescents define dating violence, as it is possible that some participants did not report experiences of sexual violence in their romantic relationship simply because they did not realize that they had experienced it; it could also happen that others think that the use of some aggression or some pressure on a romantic partner is acceptable [54,55]. Finally, and in accordance with other authors, it would be of interest to analyze what happens in other populations, apart from the large group of heterosexual adolescents attending school, like teenagers who no longer study after graduating from school and the LGBT+ community [16,17,18,26].

Although there are many open questions regarding the object of this study, this work highlights the importance of addressing teen dating violence, given the potential that such behaviors will likely persist in adult dating relationships, as indicated in previous studies [56]. A limited number of studies have measured how false beliefs and poor interpersonal skills affect sexual violence among peers, both committed and suffered; likewise, few studies analyze males and females as potential aggressors and victims, especially in adolescent couples. This double-pronged approach is relevant, at least in the Spanish context, as preventive programs with adolescents tend to work with mixed groups.

The results of this study are, in our opinion, a reflection of the poor implementation of affective sexual education in the Spanish educational system. At present, this subject is not explicitly included in the school curriculum, but is limited to a transversal approach [57]. According to Lameiras, Carrera and Rodríguez, the fact that the development of transversality has not proven to be more effective, at least in this area, is due to a lack of legislative precision, to gaps in teacher training, to a lack of interest or resources in educational centers, and to a way of understanding affective sexual education that is neither comprehensive nor highly prescriptive, and which has been limited, when put into practice, to the prevention of risks associated with sexuality (e.g., unwanted pregnancies and STI) [58]. Fostering a positive school climate has been found to be associated with reducing all types of aggression and schools must encourage victims and their peers to report incidents of dating violence so that adults can prevent the escalation of such behaviors. However, there is no doubt as to the importance of adopting a community approach to reverse dating violence and, in doing so, not only involving the adolescents themselves, but also other key socializing agents [16,17,24,26].

## Figures and Tables

**Figure 1 ijerph-17-08744-f001:**
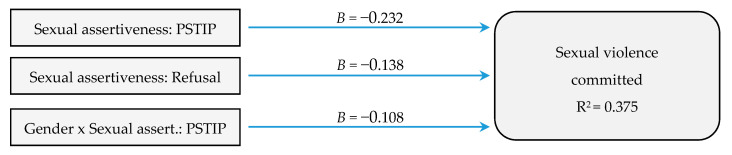
Summary of the regression analysis for sexual violence committed. (Note. PSTIP: pregnancy and sexually transmitted infections prevention).

**Figure 2 ijerph-17-08744-f002:**
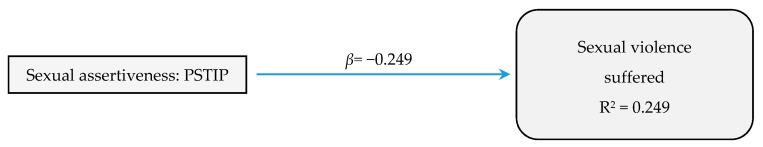
Summary of the regression analysis for sexual violence suffered. (Note. PSTIP: pregnancy and sexually transmitted infections prevention).

**Table 1 ijerph-17-08744-t001:** Descriptive statistic of the study variables and mean differences between males (M) and females (F).

N = 329	RR	Minimum		Maximum		Mean		SD		*F*	*p*
		M	F	M	F	M	F	M	F		
Myths about sexual aggression	1–7	2.23	1.60	5.67	5.13	4.02	3.66	0.49	0.72	16.53	0.00
Sexual assertiveness: Initiation	1–7	0	0.17	3.67	3.33	1.69	1.70	0.55	0.58	0.01	0.94
Sexual assertiveness: Refusal	1–7	0.33	0.67	4.00	4.00	2.31	3.00	0.89	0.87	47.44	0.00
Sexual assertiveness: PSTIP	1–7	0.83	0.17	4.00	4.00	3.21	3.48	0.90	0.85	6.78	0.01
Sexual violence committed	0–3	0	0	1.67	1.67	0.25	0.09	0.35	0.22	21.27	0.00
Sexual violence suffered	0–3	0	0	1.50	2.17	0.26	0.25	0.35	0.38	0.11	0.74

Note. RR: response rank. PSTIP: pregnancy and sexually transmitted infections prevention.

**Table 2 ijerph-17-08744-t002:** Correlations between the study variables for males and females.

	1	2	3	4	5	6
1.Myths about sexual aggression	-	−0.31 **	−0.14	−0.28 **	0.13	0.24 **
2.Sexual assertiveness: Initiation	−0.12	-	−0.16	0.05	0.05	−0.19 *
3.Sexual assertiveness: Refusal	−0.31 **	−0.11	-	0.50 **	−0.21*	−0.12
4. Sexual assert.: Pregnancy and STI prevention	−0.12	−0.18 *	0.24 **	-	−0.10	−0.28 **
5.Sexual violence committed	0.17	0.12	−0.29 **	−0.29 **	-	0.39 **
6.Sexual violence suffered	0.08	0.07	−0.18 *	−0.21 **	0.79 **	-

Note. Males’ values below diagonal; females’ values above diagonal. ** *p* < 0.01 * *p* < 0.05.

**Table 3 ijerph-17-08744-t003:** Hierarchical regression analysis for variables predicting sexual violence committed.

Model	Predictor	B	*β*	t	*p*	LT
Main effects	Gender	0.581	−0.734	−3.548	0.000	0.063
	Myths about sexual aggression	−0.036	0.089	1.604	0.110	0.877
	Sexual assertiveness: Refusal	−0.074	−0.224	−3.653	0.000	0.721
	Sexual assertiveness: PSTIP	−0.095	−0.269	−3.871	0.000	0.558
Interactions	Gender x Sexual assertiveness.: PSTIP	0.114	0.653	2.978	0.001	0.0056
		R^2^ = 0.425				
		F_5303_ = 13.380				

Note. LT: level of tolerance. PSTIP: pregnancy and sexually transmitted infections prevention.

**Table 4 ijerph-17-08744-t004:** Hierarchical regression analysis for variables predicting sexual violence suffered.

Model	Predictor	B	*β*	t	*p*	LT
Main effects	Gender	0.046	0.060	1.005	0.316	0.851
	Myths about sexual aggression	0.051	0.105	1.782	0.076	0.880
	Sexual assertiveness: Refusal	−0.027	−0.068	−1.048	0.295	0.732
	Sexual assertiveness: PSTIP	−0.089	−0.210	−3.502	0.001	0.850
		R^2^ = 0.279				
		F_4303_ = 6.395				

Note. LT: level of tolerance. PSTIP: pregnancy and sexually transmitted infections prevention.

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
