# Peer review of "Myths about Sexual Aggression, Sexual Assertiveness and Sexual Violence in Adolescent Romantic Relationships"

_ijerph, 2020, doi:10.3390/ijerph17238744_

Round 1
Reviewer 1 Report
Thank you for the invitation to review this manuscript. Research is scarce on the topic of adolescent sexual violence and rape myth acceptance.
The article Myths about Sexual Aggression, Sexual Assertiveness and Sexual Aggression in Adolescent Romantic Relationships should be considered for publication even if a few things need to be corrected or clarified. The present work has the potential to make an important contribution here.
Introduction:
In general you should check language e.g. “Sexual relations, however, do not always occur freely” is sexual relation the right term or should it be sexual encounter?
Line 38/39 shouldn’t “coital sexual assault” be a part of the examples if you refer to it afterwards?
As an interested reader I would be happy to receive further information about rape myths and the connection to sexualized violence or the prevention of sexualized violence by changing these myths E.g. Keller et al. 2017 https://journals.sagepub.com/doi/full/10.1177/0886260515586367
Methods:
Please describe the measurements and the theoretical background in more detail. Did you assess more demographic data? Sociodemographic data needs to be explained in more detail if available.
Please be more precise what variables you gathered and how you analyze them.
For example:
How many subjects you asked in total (as posteriori there were some exclusion due to the status of the relationship)?
How many subjects do you exclude due to age?
How many due to the status of relationship?
Did you experience dropouts due to other reasons? If so how many?
Are there differences in sociodemographic factors between genders?
Concerning the CADRI: which types of offenses are included? Did you pool them for the analysis? In the result section you mention at some point values for sexual coercion but refer otherwise to ‘sexual offenses committed’. This is quite confusing. So please be consistent and explain the constructs and subscales appropriately. Which factors did you include in the MANOVA?
The description of the tables is insufficient. The layout of the arrangement for mean value and standard deviation is also unusual e.g. Table 1.
Please differentiate more clearly between sexual assault and coercive behavior
Results & Discussion:
What is PSTIP? Probably pregnancy and STI prevention but the abbreviation is not explained in the text.
Summary of the main results and an interpretation of these main results should be given
“The results of this work have shown that sexual assault seems to have a significant impact on adolescent couples” on what data is this assumption based on?
“In addition, while adolescent females admitted to having committed a sexual assault of some kind, it was males who most frequently reported having committed such assaults; no statistically significant differences were found, however, between males and females in the frequency with which they reported having been assaulted.”
This sentence is confusing and misleading.
Author Response
November 9, 2020
Dear colleague,
We would like to thank you for your comments and encouragement about our paper: your constructive feedback has given us clear insights and suggestions to help us write a better manuscript. We sincerely appreciate your effort.
In addition to the list of main changes, we have undertaken an in-depth review of the writing style. We hope that the improvements made respond to your pertinent suggestions.
Once more, thank you for your conscientious and professional review.
Sincerely,
The authors
REVIEWER 1
Introduction
- All the suggestions aimed at improving this section have been made (e.g. “sexual relations” was changed to “sexual encounter” and “coital sexual violence” was added as an example of sexual victimization). [Page 1, paragraphs 1 and 2]
- Some additional references were added in order to reinforce the importance of dispelling myths and shifting attitudes away from supporting violence when it comes to prevent sexual violence: this improvement helps to highlight the importance of the objectives. [Page 2, paragraph 3 and page 3, paragraph 1]
Materials and Methods
- More sociodemographic data were provided on the sample of the study, establishing comparisons between males and females when necessary. [Page 3, paragraphs 3 and 4]
- Furthermore, more information about the sampling was included regarding, for example, the inclusion/exclusion criteria. This makes it possible to differentiate the initial sample from the final sample and how the latter was reached. [Page 3, paragraph 3]
- More detailed information was added for a better understanding of measures used (e.g. the Conflict in Adolescent Dating Relationships Inventory). [Page 4, paragraphs 3 and 4]
- Additional explanations were included on the factors considered in the MANOVA. [Page 5, paragraph 2]
- Throughout the whole paper and not only in this section, the use of the terms “sexual assault”, “sexual coercion” and “sexual violence” was reviewed: different changes were done to make a consistent use of them in order to avoid confusion.
Results
- Abbreviations are explained throughout the manuscript. [e.g. Page 9, Table 1]
- All tables were reviewed and modified for a better understanding. [Pages 9, 10 and 11]
Discussion
- The first paragraph was modified to explain better the aims of the study. [Page 12, paragraph 1]
- The expression “impact” was replaced by “presence”. [Page 12, paragraph 1]
- An introductory paragraph was included, as a brief summary, to highlight the main findings of the work. [Page 12, paragraph 1 and 2]
- The phrase about differences between males and females in sexual violence was rewritten to clarify its meaning. [Page 12, paragraph 1]
- Grammatical errors and misspellings have been corrected. In the same way, other improvements in the legibility of the text have been introduced.
- The whole manuscript was deeply reviewed, both the content and the formal aspects.

Reviewer 2 Report
Thank you for the opportunity to review your work. Overall, this is a strong manuscript that warrants publication. The introduction is particularly strong, including evidence and theoretical underpinnings that help us to understand sexual aggression and assertiveness in adolescent romantic relationships but also the gaps that emphasize the importance of the current study. Further, authors conducted appropriate analyses with a relatively large sample of adolescents and acknowledge most of their limitations in the future direction section of the discussion. I believe this work has important implications for future research and education, which are also highlighted in the discussion. Please see below for a few minor recommended revisions:
Introduction:
- Please add citations for the link between interpersonal skills and dating violence (line 91).
Materials and Methods:
- Please clarify whether adolescents were asked to sign consent again when reconfirming their consent in the survey (Lines 144-145).
- Please clarify how the question in lines 150-151 “whether they had ever been involved as perpetrators or victims in an unwanted sexual situation with their current partner” differed from questions on “sexual assault committed and suffered” in line 149.
Results:
- Minimum and maximum are spelled incorrectly in Table 1.
- In section 3.2, you indicate in the second sentence the significant correlations and then go on to describe multiple other correlations without stating whether or not they were significant. I see that significance is indicated for some of the analyses in Table 2, but perhaps a blanket statement indicating the following correlations did not reach statistical significance would allow for better transparency of the findings.
Discussion:
- The first sentence suggests sexual assault has a significant impact on the adolescent couples, but impact on the couples was not analyzed in the present study. Further, this discussion section lacks a lay summary of the findings (with the exception of myths, which is summarized I paragraph 2) and would be strengthened by including a few sentences in the first paragraph summarizing your results more explicitly.
- I appreciate the thorough and thoughtful suggestions for future directions in the discussion section, especially the importance of examining couples as a unit and minority groups such as LGBTQ youth.
- I also really appreciate the final paragraph on the importance of sexual education and fostering a positive school climate, as this gives immediate clinical implications from this work.
- In the measures section, there are no measures that analyze whether someone may “think” they were too forceful or pushy, for adolescents who are perhaps fearful to admit even to themselves that something they did was coercive or even considered assault. Please address this as a limitation or perhaps in your future direction section.
Author Response
November 9, 2020
Dear colleague,
We would like to thank you for your comments and encouragement about our paper: your constructive feedback has given us clear insights and suggestions to help us write a better manuscript. We sincerely appreciate your effort.
In addition to the list of main changes, we have undertaken an in-depth review of the writing style. We hope that the improvements made respond to your pertinent suggestions.
Once more, thank you for your conscientious and professional review.
Sincerely,
The authors
REVIEWER 2
Introduction
- Some additional references were added in order to reinforce the link between interpersonal skills and dating violence. [Page 2, paragraph 4]
Materials and Methods
- It was clarified the procedure whether adolescents reconfirmed their consent in taking part in the survey. [Page 4, paragraph 2]
- Regarding the question about whether they had ever been involved as perpetrators or victims in an unwanted sexual situation, a better explanation was included in order to clarify the aim of this question in our study. [Page 4, paragraph 2]
Results
- Minimum and maximum spelling were corrected in Table 1. [Page 9]
- The section 3.2 was rewritten in order to clarify the obtained results, which ones were for both genders and which ones were only for males or females. [Page 9, paragraphs 4, 5 and 6]
Discussion
- The first paragraph was modified to explain better the aims of the study. [Page 12]
- An introductory paragraph was included, as a brief summary, to highlight the main findings of the work. [Page 12, paragraphs 1 and 2].
- A reflection on the importance of considering in future studies what adolescents understand by dating violence was incorporated. Likewise, two bibliographical references were introduced on the importance of this issue. [Page 13, paragraph 1]
- Grammatical errors and misspellings have been corrected. In the same way, other improvements in the legibility of the text have been introduced.
- The whole manuscript was deeply reviewed, both the content and the formal aspects.
